# Using the Nematode, *Steinernema carpocapsae*, to Control Peachtree Borer (*Synanthedon exitiosa*): Optimization of Application Rates and Secondary Benefits in Control of Root-Feeding Weevils

Colin Wong [1], Camila Oliveira-Hofman [1,†], Brett R. Blaauw [2], Dario Chavez [3], Ganpati Jagdale [4], Russell F. Mizell III [5] and David Shapiro-Ilan [1,*]

1 USDA-ARS, Southeastern Fruit and Tree Nut Research Laboratory, Byron, GA 31008, USA
2 Department of Entomology, University of Georgia, Athens, GA 30601, USA
3 Department of Horticulture, University of Georgia, Griffin, GA 30223, USA
4 Department of Plant Pathology, University of Georgia, Athens, GA 30601, USA
5 Department of Entomology, University of Florida, Quincy, FL 32351, USA
* Correspondence: david.shapiro@usda.gov; Tel.: +1-(478)-956-6444
† Current address: RNAissance Ag., St. Louis, MO 63132, USA.

**Abstract:** The peachtree borer, *Synanthedon exitiosa* (Say) (Lepidoptera: Sesiidae), is a major pest of stone fruits including the peach *Prunus persica* (L.) Batsch. The entomopathogenic nematode, *Steinernema carpocapsae*, was previously shown to be an effective tool for controlling *S. exitiosa*. In orchards where irrigation is not available, a sprayable gel (Barricade®) can be used to maintain soil moisture which can facilitate nematode efficacy. However, rates of nematode and Barricade® application had not been optimized for their maximum economic and biocontrol efficiency. Therefore, our objective was to compare rates of *S. carpocapsae* and Barricade® in field trials. Nematodes were tested at per-tree application rates of 1.5 million, 1 million and 0.5 million infective juveniles. The sprayable gel was used at two rates, 4% and 2%. A reduction in the used nematodes from 1.5 million to 0.5 million per tree showed no difference in efficacy. Similarly, using the gel at half rate also did not impact the efficacy, and treatments containing nematodes controlled the *S. exitiosa* better than the chlorpyrifos control in several of the tests ($p < 0.05$). As an added benefit, the nematode treatments were also able to reduce the prevalence of weevil (Coleoptera: Curculionidae) populations as secondary pests of the peach trees. The lower rates of grower inputs will reduce costs, making the nematode biocontrol of the peachtree borer more likely to be adopted by commercial growers of peach.

**Keywords:** entomopathogenic nematodes; biocontrol; barricade® gel; root feeding weevils; *Oedophrys hilleri*; *Pseudocneorhinus bifasciatus*; *Sitona lineatus*; peachtree borer

## 1. Introduction

The peachtree borer (PTB), *Synanthedon exitiosa* (Say) (Lepidoptera: Sesiidae), is a lepidopteran pest of stone fruit trees. Johnson et al., in 2005 [1], provided a description of the basic biology of the pest with respect to the attack in orchards. Briefly, the pest can be found in orchards throughout Canada and the United States. Adult moths emerge and find mates in late-summer to mid-autumn. After mating, females oviposit above ground on substrates near the plant host including upon peach tree bark. The larvae hatch and bore into the bark of the trees above ground and tunnel their way into the root system where they remain in the root crown near the surface. Larvae will burrow further into the roots and make a pupal cell and pupate in the tunnel [1]. Once the larvae have made their way into the tree, they are relatively protected from conventional pesticide sprays, so there is a narrow window wherein adults, eggs or neonate larvae are accessible for a trunk



or soil application of pesticide [2]. Additionally, the phasing out of the organophosphate insecticide, chlorpyrifos, has eliminated one of the main control methods previously used against this pest [3,4].

Previous research had shown that the entomopathogenic nematode, *Steinernema carpocapsae*, can control *S. exitiosa* in orchards just as well or better than chlorpyrifos in controlling PTB [5–7]. Studies found that this was true in orchard conditions when applied in the fall to control the early instar larvae, as well as when applied as a curative measure in the spring after larvae have already been established in the roots of the host plant [7]. Moreover, a number of weevil (Coleoptera: Curculionidae) species including *Oedophrys hilleri* [8]; the two-banded Japanese weevil, *Pseudocneorhinus bifasciatus* [9]; the pea leaf weevil, *Sitona lineatus*; the vegetable weevil, *Listroderes difficilis*; or the white-fringed beetles, genus *Naupactus* spp. [10] are also pests of peach trees; however, they only occasionally cause enough damage to warrant treatment [10]. If chemical treatments or biocontrol organisms are applied to control *S. exitiosa*, they could also control the weevils and that could be considered as an added benefit. Entomopathogenic nematodes have already been shown to control other weevils in peach orchards that are not root feeders, e.g., plum curculio, *Conotrachelus nenuphar* (Herbst) [11]. Our research explores whether *S. carpocapsae* nematodes can also control the above stated weevil pests of peach as they feed on the roots as larvae alongside the *S. exitiosa*.

Normally, entomopathogenic nematodes (EPNs) should be applied with sufficient irrigation so that the nematodes do not desiccate and can move efficiently through the soil profile [12]. However, if irrigation is not available in peach orchards, the application of a sprayable gel (Barricade®) can help to maintain soil moisture and facilitate the survival and movement of EPNs and their biocontrol efficacy [6]. The Barricade® fire-blocking gel, which is not marketed as an agricultural product, has been shown to be compatible with agricultural uses of EPNs to control pests of plants [5,13] and animals [14]. The gel can be applied after application of the nematodes [6] or applied as a tank mix along with the nematodes [15]. The gel ensures that the soil maintains the high moisture required for optimal EPN migration [16,17]. The gel can also block UV radiation which can kill the nematodes while they are moving into the soil following an above-ground application [18]. The manufacturer's instructions for use specify a rate of approximately 4% in water. However, a lower concentration was tested as we did not use the material to fight fires (the manufacturer's intended use) and using a lower rate of gel would both reduce costs for the grower and make it more compatible with spray application machinery.

Compared to a chemical insecticide such as chlorpyrifos, which has been a mainstay for controlling *S. exitiosa* [4], nematodes can be expensive and have more strict storage requirements [19]. Additionally, using a sprayable gel to maintain the soil moisture required for the EPN application to be successful adds an additional cost. Therefore, this study examined the efficacy of lower rates of both the EPN and the sprayable gel (Barricade®, Hobe Sound, USA). These lower rates may show that an EPN strategy is competitive at a commercial scale when growers are trying to control *S. exitiosa* in their orchards.

## 2. Materials and Methods

### 2.1. Nematodes

The entomopathogenic nematodes, *S. carpocapsae,* used in this study were reared at the Byron USDA laboratory and also obtained from BASF corporation (Ludwigshafen, Germany) as their Millennium® product. Nematodes used at the Byron site were from the USDA laboratory, whereas commercial nematodes from BASF were used at the other two sites. The nematodes reared in the USDA laboratory used in vivo procedures, whereas those produced by BASF used liquid fermentation [20]. The BASF nematodes and USDA-reared nematodes were applied in separate experiments but used the same *S. carpocapsae* strain, i.e., the "All" strain, which was originally isolated in Georgia, USA. Infective juveniles (IJs) were examined under the microscope to ensure that their viability was at least 90%

and they were counted to get accurate dilutions. Nematodes were used within 2 weeks of receipt.

### 2.2. Variations on the Rate of Nematodes Used

Nematode treatments were applied each year in late September or early October to target young PTB larvae shortly after hatching from eggs. Nematodes were applied with a handgun applicator. A positive (chemical) control included Chlorpyrifos (Lorsban 4E) at a rate of 2.33 L per hectare within the range on the label. A negative (untreated) control was run at each site consisting of no application of nematodes, gel or pesticide, and normal irrigation. Treatments were grouped into replicates of five trees and were blocked using a randomized complete block design. The rates of materials used and years applied for each field location can be found in Table 1.

**Table 1.** Experimental setup across different field locations. The 'years treated' column lists the years in which a fall treatment was made, and in which data were collected the following spring. The weevils target is referring to all root-feeding weevils as secondary pests of peach.

| Location | Conditions | Treatment Year (s) | Million IJs/Tree | Barricade® Gel Rate | Target |
|---|---|---|---|---|---|
| Fayetteville | Organic | 2018 | 1.5 | - | *S. exitiosa* |
| Fort Valley Area A | Conventional | 2018, 2019 | 0.5, 1.5 | 2% | *S. exitiosa* |
| Fort Valley Area B | Conventional | 2018, 2019 | 1.0 | 2%, 4% | *S. exitiosa* |
| Byron | Conventional | 2018 | 1.5 | 4% | Weevils |

A second, curative treatment of nematodes was applied in the spring to trees that showed signs of infestation; however, this was after damage measurements were recorded. Damage measurements were recorded in the spring (mid-April) as the percentage of trees within each five-tree replicate that showed any signs of infestation.

Trees with signs of damage were categorized as infested. Signs of PTB damage were assessed following the methods described by Shapiro-Ilan et al. in 2015 and 2016 [6,7], briefly: soil was removed away from the tree's trunk and roots to 12 cm in depth. The excavated roots were probed and visually assessed to evaluate potential damage such as larval galleries, pupal cells, frass and wounds in the bark.

An organic orchard in Fayetteville, Fayette County, Georgia, was used to test the full rate of 1.5 million IJs per tree; no gel treatment of Barricade® was applied. The Fayetteville location only had enough PTB attacks in the year 2019 to assess the efficacy of treatments. The experiment consisted of four replicates of this treatment and the negative control, for a total of 20 trees per treatment.

A conventional commercial peach orchard in Fort Valley, Peach County, Georgia was used from 2018 to 2020. The Fort Valley site included two years of repeated treatments. The Fort Valley site was split into two areas and area A (Table 1) was used to test a high rate of nematodes (1.5 million IJs per tree) and a low rate of nematodes (0.5 million IJs per tree). These treatments included a 2% Barricade® gel cover, and chemical and untreated control treatments were also performed. These four treatments included four replicates with 20 trees each, resulting in 80 trees per treatment.

### 2.3. Variations in the Rate of Barricade® Gel Used

At the Fort Valley site, additional blocks (Table 1, Fort Valley Area B) consisting of treatments with 1 million IJs per tree with either a full (4%) or half (2%) rate of Barricade® gel cover were put in place. The Barricade® gel is measured as a percentage by mass and mixed in-tank with the nematodes. The gel in the tank along with the nematodes is constantly agitated and sprayed at 60 PSI (~400 kPa). The applications are made to the trunk of the tree near the ground and the nematodes can move out of the gel or are pushed out after rainfall. A set of chemical and untreated controls were also included. These

treatments used five trees per replicate with four replicates, resulting in 20 trees used per treatment. This area of trees was distinct from the blocks used for the nematode rate tests, and as such, the 1.0 million IJs per tree rate is not compared to the other nematode rates for statistical analysis.

### 2.4. Effects of Treatments on Weevil Secondary Pests

The research orchard at the USDA, ARS, Southeastern Fruit and Tree Nut Laboratory in Byron, Georgia (coordinates: 32.657, −83.742) was used to study whether the treatments for PTB also had an effect on secondary pests, namely root-feeding weevils. Following the nematode application at the Byron site in the fall of 2018, Circle traps were placed around the trunk of each tree and the traps were checked weekly from May 2019 to May 2020. The Circle traps consist of a mesh (1.5 mm gap size) cone facing the bottom of the tree with a removable screw-top plastic container at the top. Circle traps were manufactured by hand. Adult weevils were identified to one of the following species: *Oedophrys hilleri*, *Pseudocneorhinus bifasciatus*, *Sitona lineatus*, *Listroderes difficilis*, the genus *Naupactus* spp., or as "other". Weevil totals were collected and determined for the different treatments. The nematode treatment was applied at the full input treatment (1.5 million IJs and 4% gel), and chemical and untreated controls were also included. These tests used five tree replicates with four replicates per treatment, using a total of 60 trees.

### 2.5. Statistics

An Analysis of Variance (ANOVA) was used to determine whether any treatments were statistically different from one another (SAS Version 9.4, 2002). A general linear model using the least squares means was then used to calculate pairwise differences post hoc, using an alpha value of 5% to determine significance.

## 3. Results

### 3.1. Entomopathogenic Nematode-Only Treatments

In the Fayetteville site, the trees with EPN treatments showed no signs of infestation by moths, and this was significantly lower than in the untreated control treatment (Figure 1. $p < 0.05$).

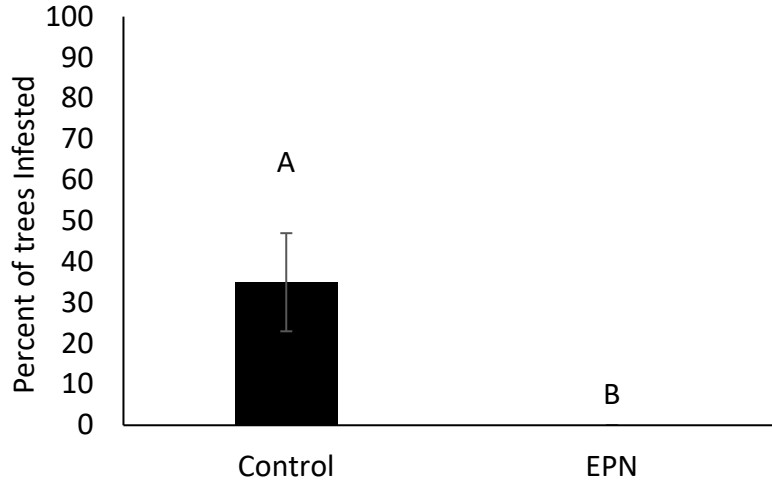

**Figure 1.** Percentage of peach trees infested with peachtree borers following treatment of the entomopathogenic nematode (EPN), *Steinernema carpocapsae*. Control = no nematodes applied. The application was made in an organic peach orchard in fall 2018 with a reapplication and assessment in spring 2019. Different letters above bars indicate significant differences ($p < 0.05$).

### 3.2. Variations on the Rate of Nematodes Used

The EPN rate comparison showed that over both years at the Fort Valley site, the two application rates were matched in efficacy. Figure 2 shows that in the first year

there were significant differences between the treatments ($F_{3,12}$ = 11.29, $p$ < 0.005). Post hoc testing showed that both EPN treatments (0.5 and 1.5 million IJs concentrations) significantly reduced the percentage of infected trees compared to the untreated control ($p$ < 0.005). The high concentration also caused significantly reduced infections compared to the chlorpyrifos treatment (0.5 million $p$ = 0.13, 1.5 million $p$ = 0.005). The second year, there were also significant differences between the treatments ($F_{3,12}$ = 4, $p$ = 0.035), and the post hoc tests revealed that the 0.5 million and 1.5 million IJs-per-tree treatments and the chlorpyrifos were all significantly different from the untreated control (and not different from each other, Figure 3) ($p$ = 0.012, $p$ = 0.026, and $p$ = 0.012, respectively).

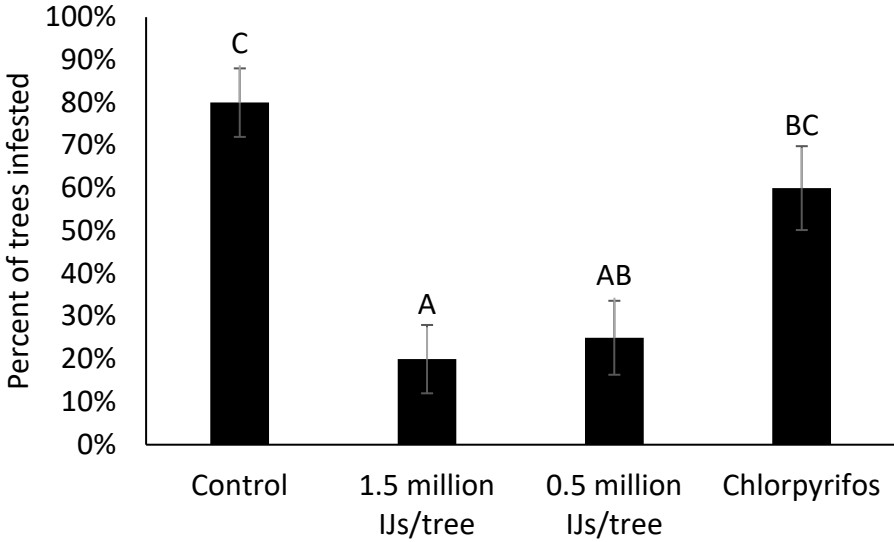

**Figure 2.** Percentage of peach trees infested with peachtree borers in 2019. Control = no nematodes applied, IJs = infective juveniles. The application was made in a conventional peach orchard in fall 2018 with a reapplication and assessment in spring 2019. Different letters above bars indicate significant differences ($p$ < 0.05).

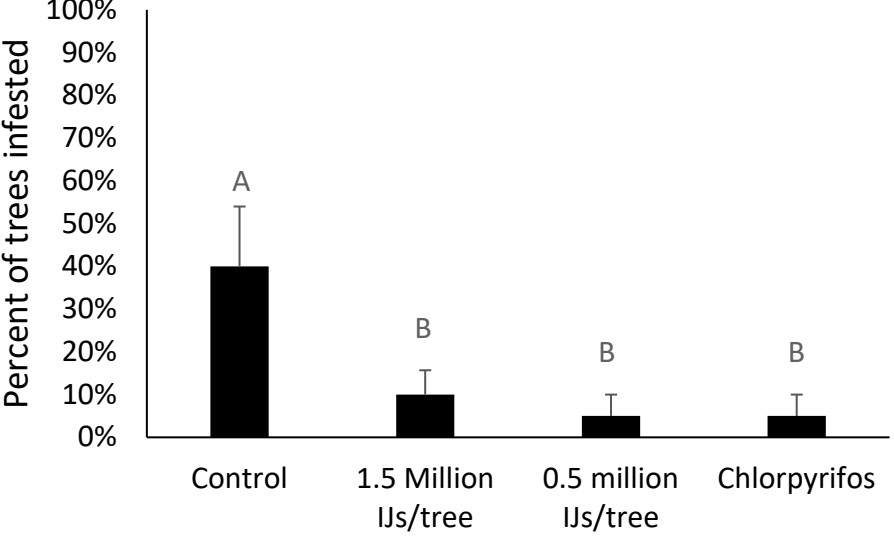

**Figure 3.** Percentage of peach trees infested with peachtree borers in 2020. Control = no nematodes applied, IJs = infective juveniles. The application was made in a conventional peach orchard in fall 2019 with a reapplication and assessment in spring 2020. Different letters above bars indicate significant differences ($p$ < 0.05).

### 3.3. Variations in the Rate of Barricade® Gel Used

The Barricade® rate comparison at the Fort Valley site revealed no difference between using the full rate or cutting the rate of application in half. In the first year, shown in Figure 4, the two Barricade® treatments had significantly lower percentages of infestation than both the untreated and chemical controls ($F_{3,12}$ = 10.57, $p < 0.005$ overall; $p \leq 0.005$ for all pairs). The two Barricade® rates were not different from each other ($p = 0.74$, Figure 5). In the second year, the two EPN-containing Barricade® treatments completely eliminated damage from potential infections in the treated trees, and were significantly lower than the controls (Figure 5. $F_{3,12}$ = 4.3, $p = 0.028$). The chemical control had significantly lower infestation rates than the untreated control that second year ($p = 0.025$).

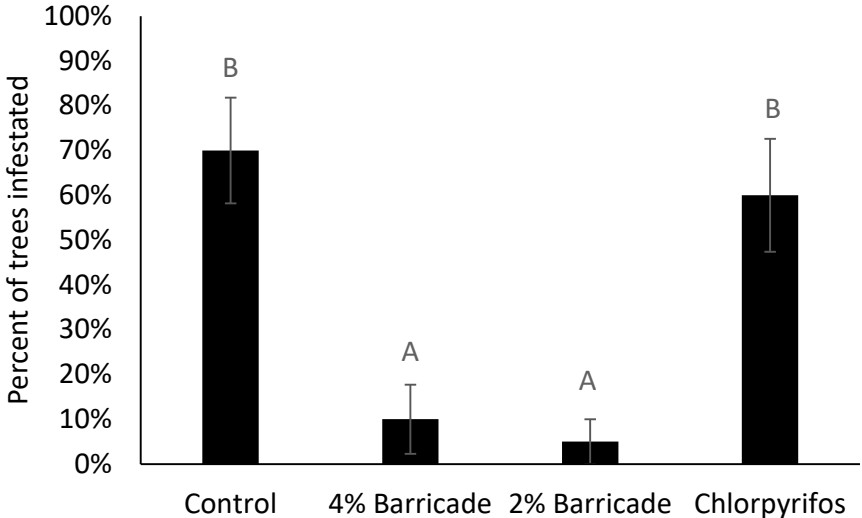

**Figure 4.** Percentage of peach trees infested with peachtree borers, comparing rates of Barricade® treatments (with *Steinernema carpocapsae* nematodes, 2019). Control = no nematodes applied. All treatments used 1 million infective juveniles per tree. The application was made in a conventional peach orchard in fall 2018 with a reapplication and assessment in spring 2019. Different letters above bars indicate significant differences ($p < 0.05$).

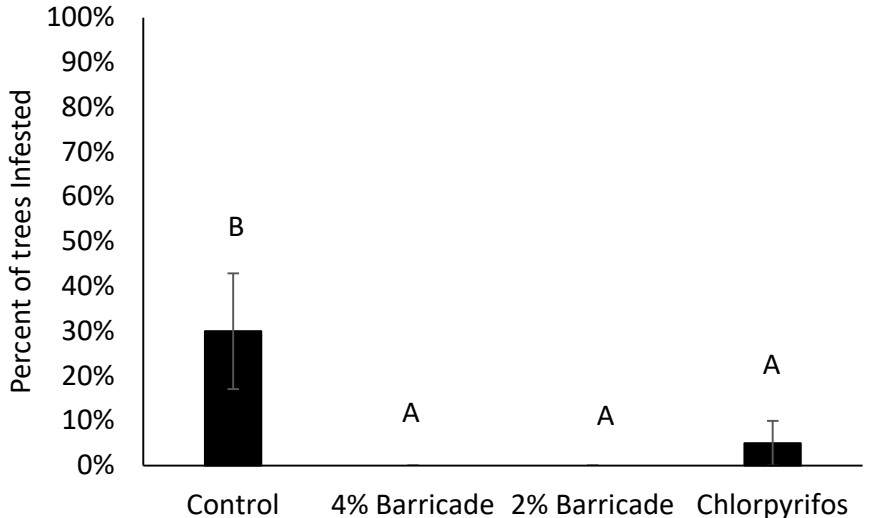

**Figure 5.** Percentage of peach trees infested with peachtree borers, comparing rates of Barricade® treatments (with *Steinernema carpocapsae* nematodes, 2020). Control = no nematodes applied. All treatments used 1 million infective juveniles per tree. The application was made in a conventional peach orchard in fall 2019 with a reapplication and assessment in spring 2020. Different letters above bars indicate significant differences ($p < 0.05$).

### 3.4. Effects of Treatments on Weevil Secondary Pests

We found that the application of EPNs also reduced the population of secondary pests, including different species of root-feeding weevils. For example, the nematode treatments reduced the total number of emerging adult weevils, while the chemical control did not ($p < 0.05$) (Figure 6). The total catch (2480 adult weevils) were separated by identity (Figure 7). *Oedophrys hilleri* made up the greatest proportion of beetles caught at 41.9%. *Pseudocneorhinus bifasciatus* was the next most abundant with 17.9% of the collected adults. The species *Sitona. lineatus* made up 15.3%; the fuller rose beetles *Naupactus godmanni* made up 11.2%, while the rest of the genus of white-fringed beetles, *Naupactus* spp., made up 3.7%; the species *Listroderes. difficilis* accounted for 6.5%; and the remaining weevils (3.7%) were pooled as "other".

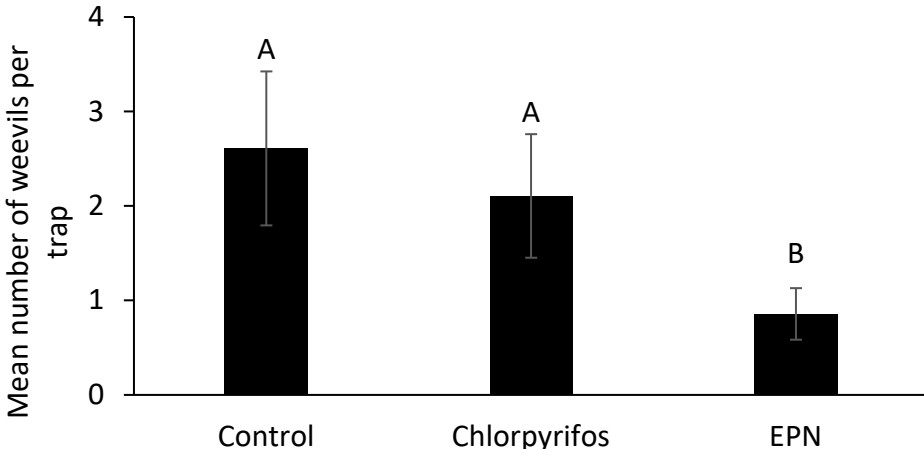

**Figure 6.** Average weevil catch per week. EPN = treatment containing entomopathogenic nematodes, *Steinernema carpocapsae*. A combination of all species of weevil caught from May 2019 to May 2020. Different letters at the top of bars indicate significant differences ($p < 0.05$).

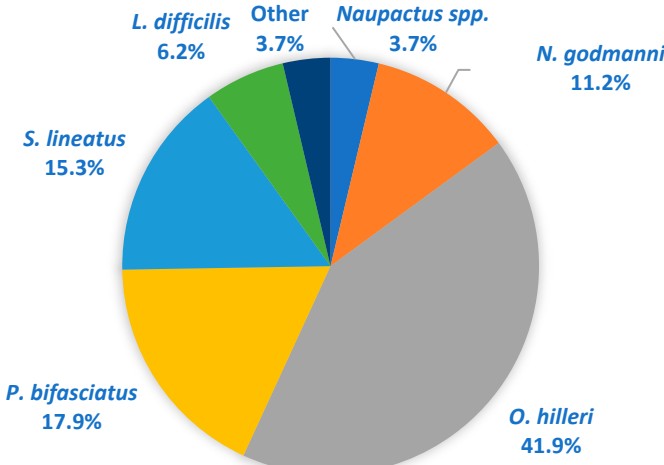

**Figure 7.** Weevil species composition. The weevils found on peach roots caught from May 2019 to May 2020 (2480 total adult weevils), with common clades identified.

## 4. Discussion and Conclusions

The nematode treatments were effective in controlling *S. exitiosa* in the field in both conventional and organic field sites. These results align with the previous research [5–7] and corroborated the effectiveness of *S. carpocapsae* against this pest as was shown in previous years. The results at the Fayette County site are the first reporting of peachtree borer control with *S. carpocapsae* in an organic setting. Damage from PTB was entirely eliminated at the Fayette County site from trees treated with EPNs; however, the number

of damaged trees in the control treatment was also low. The second year of planned trials in the organic orchard were unable to be completed as the PTB infestation was too low in the location. The Fayette County data support the use of EPNs for biocontrol in peaches but longer studies using EPNs as the sole control method are still needed.

Conceivably, other nematode species besides *S. carpocapsae* may be effective in field trials. The nematode species and strain we chose were based on the efficacy observed in previous studies in peach orchards [5–7]. Moreover, laboratory trials challenging *S. pictipes* with six different species of EPN indicated *S. carpocapsae* (All) to be the most efficacious [21]. Specifically, in this laboratory trial, *S. carpocapsae* was more virulent towards the lesser peachtree borer (close relative to the target of this research) than three species of *Heterorhabditis* EPN as well *as Steinernema feltiae* and *Steinernema riobrave* [21]. Field trials in Arkansas found that *S. carpocapsae* had the greatest control of the sesiid pest *Pennisetia marginata*, compared to H. bacteriophora and S. feltiae [22].

Chlorpyrifos had been a mainstay of PTB control for decades but is no longer registered for this purpose [4,23]. The nematode treatments performed as well as or better than the chlorpyrifos-containing insecticide treatments. However, chlorpyrifos showed relatively low control in our treatments due to late applications (late September and early October) after larvae of the insect had already hatched and begun to bore into the trees. This shows that biocontrol with EPNs is more versatile than chemical control against this root-dwelling pest, as the nematodes can move through the substrate and follow the insects into the roots. Entomopathogenic nematodes have been used in this way for other pests with cryptic behavior [24,25].

Mating disruption has also been explored as an alternative approach to *S. exitiosa* control [26]. Mating disruption has been effective in trials; however, the technique can be expensive and requires deployment over large areas [26,27]. EPNs can be an alternative tool for orchards of a smaller size or as a supplementary tool if there is a local failure in a mating disruption strategy.

The Fort Valley orchard had consistent data from tests repeated over two years showing that the treatments were still effective with reduced inputs. The reduced rate of nematodes was as effective as the full rate. Lowering the required concentration of live IJs to one-third of the previously used quantities may make the use of EPNs much more competitive as a biocontrol alternative. The lower application rate and the potential to target only around the base of trees makes this use of EPNs potentially much cheaper than in row crops, which can require a minimum of $2.5 \times 10^9$ IJs/ha [12].

The Barricade® gel was effective as a treatment to help the nematodes establish themselves at the full, 4% rate and no-less-so at the half rate (2%) treatment. The application of the gel along with the nematodes reduced the number of treatments performed by the growers. When used against the lesser peachtree borer (*Synanthedon pictipes)*, the success of both the 2% gel and *S. carpocapsae* IJs was in agreement with the previous studies [15]. Including the nematodes in a flowable gel for pre-sale storage has been explored but was less effective in promoting the storage longevity of the nematodes than other methods of shelf-life extension [28]. Beads or capsules of polysaccharides derived from algae have been used to encapsulate EPNs and deploy them against corn rootworm, *Diabrotica virgifera*, a soil pest [29]. Alginate capsules are sometimes described as gels but are not sprayable and are designed for different targets than a sprayable gel. Alternative gels that may be more effective and less expensive or a further reduction in the rate of the Barricade® gel formulation should be explored to find a formulation that can be mixed in a single tank and be applied via the diverse equipment currently used by growers.

The weevil tests at the Byron site found that with a single application of nematodes a year, the number of adult weevils emerging from the base of the tree was reduced. These weevils can cause added stress to a tree through defoliation as well as larval feeding on the roots, and the reduction in this burden will likely help trees to weather other stressors [10]. This could allow for the trees to indirectly fend off other pests and diseases, but the entomopathogenic nematodes themselves could have a direct impact on several additional

stressors. EPNs and their bacterial metabolites can reduce damage from plant parasitic nematodes in greenhouses and row crops [30,31]. Additionally, the symbiotic bacteria found in EPNs can reduce damage from phytopathogenic fungi such as the *Armillaria* sp. fungi that cause root rot disease [32]. Taken together, these marginal benefits should enhance the overall health of the crop.

The ability for EPNs to actively move through the soil and seek out cryptic life stages is a key advantage over conventional insecticides for the control of insect pests such as *S. exitiosa* and root-feeding weevils. The findings of this study support bringing EPNs into orchard management practices under an integrated framework and suggest that this can be achieved at a lower cost than previously examined. One company recently indicated an approximate cost of USD 25.00 per acre (~USD 61.88 per hectare), which is comparable to many chemical insecticides even when factoring in the product and application costs for Barricade®. The low cost, high levels of efficacy and secondary benefits (such as root weevil suppression) make the EPN option for PTB control attractive and economically viable.

Future studies should inquire whether the control of *S. exitiosa* can be achieved for multiple generations with a single application of EPNs, thereby further improving their economic competitiveness. Strains of nematodes that can persist in the soil for long periods, allowing for a long-term control with fewer EPN applications [12,33] and without the problem of dangerous residues that accompanies long-lasting chemical insecticides, have been found and developed. The prospect of *S. caprocapsae* or other strains of nematodes remaining virulent in the soil for multiple seasons of PTB offspring could further encourage the use of biocontrol in this system.

**Author Contributions:** Conceptualization, D.S.-I., C.O.-H., B.R.B., D.C., G.J. and R.F.M.III; methodology, D.S.-I., C.O.-H., B.R.B., D.C., G.J. and R.F.M.III; formal analysis, C.W., C.O.-H. and D.S.-I.; investigation, D.S.-I., C.O.-H. and B.R.B.; writing—original draft preparation, C.W.; writing—review and editing, D.S.-I., C.O.-H., B.R.B., D.C., G.J. and R.F.M.III; supervision, D.S.-I.; project administration, D.S.-I. and C.O.-H.; funding acquisition, D.S.-I., B.R.B., D.C., G.J. and R.F.M.III. All authors have read and agreed to the published version of the manuscript.

**Funding:** This research was funded by USDA-NIFA-SARE (Sustainable Agriculture Research and Education) Award Number LS18-298.

**Data Availability Statement:** The data presented in this study are available upon request from the corresponding author.

**Acknowledgments:** We would like to thank Stacy Bryd and Kate Anderson for their technical assistance. We would like to thank the BASF corporation for the provision of commercial nematode formulations.

**Conflicts of Interest:** The authors declare no conflict of interest.

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
