# Peer review of "Using the Nematode, Steinernema carpocapsae, to Control Peachtree Borer (Synanthedon exitiosa): Optimization of Application Rates and Secondary Benefits in Control of Root-Feeding Weevils"

_agronomy, doi:10.3390/agronomy12112689_

Round 1
Reviewer 1 Report
These works are interesting but their presentation must be improved. The experimental set-up must be precised. Recording or counting the PTB pests must be explained. A picture of the different PTB stages should be given.
In details, the following corrections must be carried out.
Change the title :
Control of the peachtree borer (Synanthedon exitiosa) with the nematode Steinernema carpocapsae: optimization of application rates and secondary benefits in control of root-feeding weevils
Abstract:
Remove: “(full rate on the label)”
Better explain that these are application rates
Remove: “(the previously tested rate)”
Reformulate this sentence: “With lower rates of 27 grower inputs being required this form of biocontrol will be more feasible to be adopted by commercial growers of peach.”
Weevil: which species
Line 36: complete with the climatic zone and type
Lines 72-76: cut this long sentence in wow new sentences
Line 86: S. carpocapsae in italics
Line 86: what does mean “(“All” strain)”? Is it the name of the strain?
Line 86-92: Is there more specific information on these 2 nematodes populations used in these works?
Line 98; please use the International System of Units, change pints and acre for ISU volume units
Line 103: change spring for Spring
Section 2.2. Variations on the rate of nematodes used
It is not clear how many experimental set-ups were used? It seems that there are 3 experiments. Make it more understandable at the beginning of the section. I understand that the gel Barricade was only used in the 3rd experiment
Precise how Barricade is applied.
Section: 2.3. Variations in the rate of Barricade® gel used
Please merge this section with 2.2. and organize in a clear manner the four experiments
Line 130: precise the model and provider of the emergence traps
Line 132 : give the full names for the genera and italicize the weevils names: “O. hilleri, P. bifasciatus, S. lineatus, L. difficilis”
Same for the genus name “Naupactus”
Line 188: italicize “ Steinernema carpocapsae”
2. Materials and Methods in relation with figures from results.
You must add the information on how you record the PTB presence.
For instance, in Figure 1., the legend speaks of “percent infested”. How do you record the presence of the PTBs or count the PTBs on the trees”
Use the same legends in Figure 1 as in Fig,2, 3,4 and 5, i.e. replace “percent infested” in fig 1 by “percent of trees infested”.
Fig.6 Please reframe the figure.
Line 283: use ISU surface units instead of “acre”
Bibliography: uniformize the style and fonts. Enrich the bibliography and generally the paper, and the discussion, with articles, which are not only from Shapiro-Ilan, one of the co-author and the corresponding author. This makes to many self-citations.
Line 314 : italicize “Synanthedon exitiosa” and “Synanthedon pictipes”
Line 319-320: italicize “Synanthedon exitiosa”
Line 324: same for “Oedophrys hilleri”
Line 326: same for “Pseudocneorhinus bifasciatus”
Line 336, 339, : same for “Steinernema carpocapsae”
Line 359: correct ” etomopathiogenic”
Line 362; italicize “Meloidogyne javanica”
Line 363-364: same for “Xenorhabdus” and “Photorhabdus”
Author Response
Thank you,
Please see attachment.

Reviewer 2 Report
General comments
The manuscript “Steinernema carpocapsae to control peachtree borer (Synanthedon exitiosa): optimization of application rates and secondary 3 benefits in control of root-feeding weevils” examines the efficacy of different concentrations of the entomopathogenic nematode (EPN) Steinernema carpocapsae and the sprayable gel (Barricade®) against the pest Synanthedon exitiosa.
This is an interesting topic because maintaining soil moisture facilitates the survival and movement of EPNs for better biocontrol efficacy.
I’m not a native English speaker but the manuscript is clear and fluid to read, although some minor corrections should be made in the text, some parts of the article should be clarified.
Materials and methods:
ž Pag.2, line 86: Please write S. carpocapsae in italic and check all scientific names in the text (e.g.: pag.3, line 132).
ž Pag.3 line 107 – 112 – 128: I suggest entering the geographic coordinates of the three experimental sites;
ž Pag. 3, lines 120-124: Why the authors didn’t test the treatment with 1 million IJs 120
ž per tree without Barricade? It would have been an important data to evaluate the effective effectiveness of the gel with that concentration of nematodes
ž Pag. 3, line 121: sometimes Barricade has ®, sometimes not… Please check all manuscript and uniform it.
Results
ž Pag.4, line 154: Please add in which site.
ž Pag.4, line 158 (P <0.005)
ž Pag.4, lines 175-184: the percentages in the text do not match with those shown in the figure 7
Figure
ž Fig.6: Please replace the figure because it is cut out.
Discussion and Conclusion
· Pag.8, line 225-226: I think it isn’t correct to write ‘The nematode treatments…” because the authors have only tried one species, so I suggest changing to “Treatments with S. carpocapsae”, and to explain better why not other EPNs such as Heterorhabditis bacteriophora or S. feltiae have also considered for the experiments. Maybe they have already been tested in previous works? Please add this part, also because in line 278 the authors write of active movement of nematodes, but it is known that S. carpocapsae is not among the most mobile EPNs.
References
I think the references in the text are not in the correct order since after 7 there are 11 - 12 and then 8. Please review the journal guidelines and put the cited articles in the correct order.
Author Response
Thank you,
Please see attachment.

Reviewer 3 Report
The experimental design is not consistent with the data shown. There are major discrepancies in the results and in how they were represented in the figures.
The captions are uninformative; comparing the information in Table 1 with the figures is impossible without looking up the relevant information in the text.
If the pesticide used is not registered for use on this pest, why is it used as a control and at a time when it is not effective?
The figures are inaccurate, the axes dissimilar, there are typos.
Figure 7 is useless as it is not relevant to the purpose of the paper.
The work could not be accepted in this form. It must be deeply rewritten and revised.
Reviewer 4 Report
The work needs drastic revision before submission.
The data obtained in this work come from assays conducted in previous years, but there seems to be a lack of consistent and continuous logic in the experimentation.
Why was Barricade not used in the assays of 2018 at Fayetteville?
Why is there no data on 1 million IJ/tree. They are missing in Figs. 2 and 3 and were used in Figs. 5 and 6.
Graph figures: Uniform the scale on the Y axis for all graphs and correct the mistakes.
Pesticide treatment: the authors state that it is not effective, so the comparison is favorable to EPNs, but in Fig. 3 the treatment with Chlorpyrifos gives an infestation rate of about 5%, so it seems to be highly effective.
Table 1: The table must be corrected and integrated with more information about treatments and target pests.
There are discrepancies between the various treatments carried out in different locations. For example, why was only one EPN concentration tested in Byron?
Same question for the use of Barricade, in some locations it was not applied in others at different concentrations. In addition, the regions where the treatments shown in Table 1 were carried out are not specified in the graphs.
Thus, it is difficult to compare table and graphs.
To interpret the results, it is mandatory to read the Materials and Methods, so it is not easy and user-friendly.
Fig. 1 - Add “trees” on the Y axis. EPNs concentration applied?
Figure 7 - is not relevant to the context of this work.
Author Response
Thank you,
Please see attachment.

Round 2
Reviewer 1 Report
The manuscript has been improved but I regret very much that corrections have not been made visible in the new version of the manuscript. This is normally asked in the revision procedure. Please ask them to send the manuscript with apparent corrections.
Author Response
A copy of the manuscript with changes marked (from both rounds of revisions) is attached in the supplemental documents.
An additional citation was included (Kim et al. 2021) to bring up to date discussion around alternative formulations.
Reviewer 2 Report
Accept in present form
Author Response
Thank you for your review.
Reviewer 3 Report
I suggest the Authors revise the figures to improve them, e.g. axes are missing in some figures
Author Response
Axes and borders on figures revised to be present and consistent.
Reviewer 4 Report
Please revise the graphs again; particularly, Y axes are not visible in some figures.
Author Response

(The authors gave the same response as above.)
